

# Self-stabilized Bose polarons

Richard Schmidt[1,2] and Tilman Enss[3]

**1** Max-Planck-Institute of Quantum Optics,
Hans-Kopfermann-Straße 1, 85748 Garching, Germany
**2** Munich Center for Quantum Science and Technology,
Schellingstraße 4, 80799 Munich, Germany
**3** Institut für Theoretische Physik, Universität Heidelberg, 69120 Heidelberg, Germany

## Abstract

The mobile impurity in a Bose-Einstein condensate (BEC) is a paradigmatic many-body problem. For weak interaction between the impurity and the BEC, the impurity deforms the BEC only slightly and it is well described within the Fröhlich model and the Bogoliubov approximation. For strong local attraction this standard approach, however, fails to balance the local attraction with the weak repulsion between the BEC particles and predicts an instability where an infinite number of bosons is attracted toward the impurity. Here we present a solution of the Bose polaron problem beyond the Bogoliubov approximation which includes the *local* repulsion between bosons and thereby stabilizes the Bose polaron even near and beyond the scattering resonance. We show that the Bose polaron energy remains bounded from below across the resonance and the size of the polaron dressing cloud stays finite. Our results demonstrate how the dressing cloud replaces the attractive impurity potential with an effective many-body potential that excludes binding. We find that at resonance, including the effects of boson repulsion, the polaron energy depends universally on the effective range. Moreover, while the impurity contact is strongly peaked at positive scattering length, it remains always finite. Our solution highlights how Bose polarons are self-stabilized by repulsion, providing a mechanism to understand quench dynamics and nonequilibrium time evolution at strong coupling.

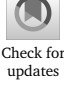
## 1 Introduction

Impurities in a Bose-Einstein condensate (BEC) exhibit a multitude of fundamental physical phenomena: the formation of quasiparticles [1], Efimov bound states [2, 3], synthetic Lamb shifts [4, 5], Casimir interactions induced by a fluctuating medium [6, 7], and quantum criticality [8]. Current experiments with ultracold atomic gases are investigating several of these effects reaching far into the strong-coupling regime [4, 8–13]. For understanding experimental observations it is thus vital to develop a theoretical model that applies to impurity systems at strong coupling and that can address both ground state and non-equilibrium phenomena.

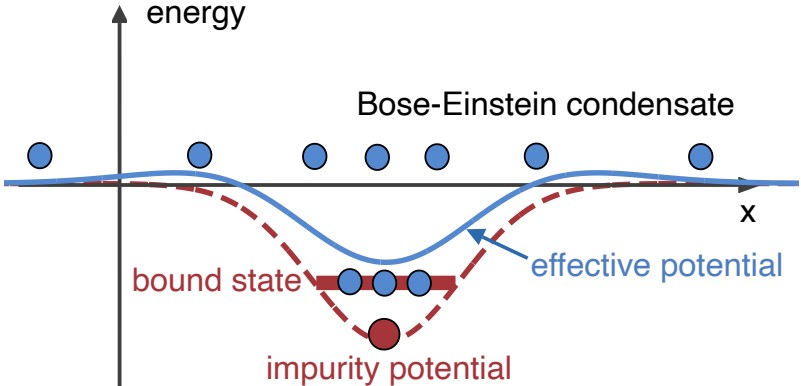

Figure 1: Illustration of the self-stabilized Bose polaron. The strong-coupling Bose polaron mimics a microtrap (red dashed line) with bound state (red bar) within the Bose-Einstein condensate. If several bosons (blue dots) occupy the bound state, boson repulsion results in a shallower effective potential seen by the remaining bosons (blue solid line) that no longer admits a bound state. The dressing cloud itself thus stabilizes the Bose polaron.

At weak interactions between the impurity and the BEC, the impurity deforms the BEC only slightly and physics is well described within the Fröhlich model in terms of long-wavelength phonon excitations [1, 14–24]. For strong local attraction, instead, the Fröhlich model is incomplete and needs to be amended by quadratic terms that absorb and re-emit phonons [25]. Importantly, these term are also required to correctly capture the formation of bound states between the impurity and bath atoms which is a crucial ingredient for the physics of polorans at strong coupling [26]. Variational wave functions based on a single phonon excitation [27, 28] are able to describe the single occupation of such a bound state. However, the bosons that make up the BEC tend to bunch, and at strong coupling it is energetically favorable to occupy the bound state multiple times leading to a gain of several times the binding energy. This process, recently observed with Rydberg atoms immersed in a BEC [29], is described by a coherent state variational ansatz [25, 26, 30, 31] that allows for an arbitrary number of excitations and strong local deformations of the condensate [32–38].

Generally, the application of the variational principle for the determination of the ground

state is only viable if the Hamiltonian is bounded from below. Under this condition a stable solution can be found, and theoretical approaches should rely only on such approximations that preserve stability of the underlying problem[1]. For the strongly interacting Bose polaron, the resulting theoretical challenge can be understood in the simple toy model illustrated in Fig. 1. Here a static attractive impurity is represented by a local potential well of finite range around the impurity. This well acts as a microtrap within the BEC. At strong coupling beyond a scattering resonance [39], the potential well (red dashed line) is deep enough to admit a bound state with energy $\varepsilon_i = -\varepsilon_B < 0$ (red bar). In absence of boson repulsion, in the many-body ground state *all* bosons would occupy the bound state and the ground-state energy $E_0 = -N\varepsilon_B$ is indeed unbounded from below in the thermodynamic limit: the whole BEC is collapsed onto the impurity.

Naturally, a local boson repulsion counteracts this process by balancing the impurity attraction [40] and thus providing a lower bound to the ground-state energy. This mechanism can be understood in terms of a single-site Bose-Hubbard model [41,42] with local repulsion $\frac{U}{2}n_i(n_i-1)$ on the impurity site $r_i$, that competes with a local attractive potential energy $-\varepsilon_B n_i$ for occupation number $n_i$. For $U > 0$ the ground state has a finite occupation $n_i \simeq 2\varepsilon_B/U$, which is nonperturbative in the strength of the interaction: the size of this polaron dressing cloud grows for weaker repulsion. It is crucial to adequately capture this repulsive effect in the theoretical description of Bose polarons.

Previous variational approaches have included the boson repulsion only at the level of the Bogoliubov approximation. Here the interaction between the Bogoliubov quasiparticles is neglected, and thus bosons in the bound state fail to generate the compensating pressure required to ensure the stability of the ground state. As a consequence this approach falsely predicts instead a dynamical instability in presence of boson repulsion toward infinite occupation of the bound state in the strong coupling regime [30,31,36]. This shows the crucial importance of including the boson repulsion beyond the Bogoliubov approximation.

In this work, we present a stable variational approach to the Bose polaron problem at strong coupling. Our approach applies to arbitrary dimension and impurity-boson scattering lengths and it provides a basis for the study of dynamical properties of Bose polarons. In this way our work complements and extends previous approaches using quantum Monte Carlo to determine ground-state properties [43], studies of the role of boson repulsion in one-dimensional systems [44–48], or a recently developed nonlocal Gross-Pitaevskii (GP) theory for nonequilibrium dynamics [40]. Moreover, previous works raised the question to which extent polaron properties are universal in the short-range limit [49,50]. In the following we study this question across the Feshbach resonance, and in particular in the regime where a bound state is supported by the impurity-bath potential, extending previous work that considered how the properties of the Bose polaron depend on the range of the interactions, both for the finite-range boson repulsion in the nonlocal GP theory [40] and for finite-range impurity potentials [42,51,52].

Specifically, we study the effect of local boson repulsion and finite-range attractive impurity potentials employing an inhomogeneous variational state that allows for large dressing clouds and strong local deformations of the BEC. In Sec. 2 we introduce the stable Bose polaron model and discuss its solution within Gross-Pitaevskii theory. In Section 3 we minimize the resulting GP energy functional and obtain the condensate wave function around the impurity. We find that even if the bare impurity potential admits a bound state, the emerging effective potential does not, thus providing a simple mechanism for the self-stabilization of Bose polarons. Section 4 presents our results for the polaron energy, the size of the polaron dressing cloud and the Tan contact across the resonance. The universality of the polaron energy is discussed in

---

[1]In this work we restrict ourselves to models of cold dilute gases that disregard deeply bound states, transitions to solid or liquid phases, as well as large cluster formation.

Sec. 5, and we show that the energy at unitarity depends universally on the effective range, as long as Efimov states can be neglected. Finally, in Sec. 6 we compare variational approaches to the Bose polaron and discuss which Hamiltonians and energy functionals can provide rigorous bounds on the ground-state energy.

## 2 Model

We consider a single impurity particle immersed in an interacting Bose gas. The combined system is described by the Hamiltonian

$$H = \frac{\hat{\boldsymbol{p}}^2}{2m_I} + \sum_i V_{IB}(\hat{\boldsymbol{x}}_{B,i} - \hat{\boldsymbol{x}}) + \sum_i \frac{\hat{\boldsymbol{P}}_{B,i}^2}{2m_B} + \sum_{i<j} V_{BB}(\hat{\boldsymbol{x}}_{B,i} - \hat{\boldsymbol{x}}_{B,j}). \tag{1}$$

Here, $\hat{\boldsymbol{p}}$ and $\hat{\boldsymbol{x}}$ denote the momentum and position of the impurity of mass $m_I$, and $\hat{\boldsymbol{p}}_{B,i}$ and $\hat{\boldsymbol{x}}_{B,i}$ characterize the bosons $i = 1, \ldots, N$ of mass $m_B$. The boson interaction $V_{BB}(\boldsymbol{x})$ is assumed to be repulsive and the impurity-boson interaction $V_{IB}(\boldsymbol{x})$ is attractive.

The coupled system of impurity and bosons is conveniently analyzed in the reference frame comoving with the impurity. This is achieved by a canonical transformation introduced by Lee, Low, and Pines [53] and elaborated on by Girardeau [15]. The transformation $S = \exp(i\hat{\boldsymbol{x}} \cdot \sum_i \hat{\boldsymbol{p}}_{B,i})$ leads to the LLP Hamiltonian

$$H_{LLP} = S^{-1}HS = \frac{(\boldsymbol{p}_0 - \sum_i \hat{\boldsymbol{p}}_{B,i})^2}{2m_I} + \sum_i V_{IB}(\hat{\boldsymbol{x}}_{B,i}) + \sum_i \frac{\hat{\boldsymbol{P}}_{B,i}^2}{2m_B} + \sum_{i<j} V_{BB}(\hat{\boldsymbol{x}}_{B,i} - \hat{\boldsymbol{x}}_{B,j}), \tag{2}$$

in which the impurity operators have been eliminated, and $\boldsymbol{p}_0$ denotes the conserved total momentum of the system. In the comoving frame, the impurity potential $V_{IB}(\boldsymbol{x})$ acts as a static external potential centred at the origin, while the kinetic term proportional to $\sim 1/m_I$ accounts for the recoil of the impurity. For bosons in the vicinity of the impurity this term leads to induced interactions between bosonic particles in addition to their inter-boson repulsion $V_{BB}(\boldsymbol{x})$.

**Homogeneous Bogoliubov Approximation.**—In order to appreciate the importance of the adequate inclusion of boson repulsion we briefly review approximations to the model (1) that are frequently applied to the study of the Bose polaron problem. In the formalism of second quantization Eq. (1) reads

$$\hat{H} = \sum_{\mathbf{p}} \frac{\mathbf{p}^2}{2m_I} \hat{d}_{\mathbf{p}}^\dagger \hat{d}_{\mathbf{p}} + \sum_{\mathbf{p}} \frac{\mathbf{p}^2}{2m_B} \hat{a}_{\mathbf{p}}^\dagger \hat{a}_{\mathbf{p}} + \frac{1}{\mathcal{V}} \sum_{\mathbf{k}\mathbf{k}'\mathbf{q}} V_{IB}(\mathbf{q}) \hat{d}_{\mathbf{k}'+\mathbf{q}}^\dagger \hat{d}_{\mathbf{k}'} \hat{a}_{\mathbf{k}-\mathbf{q}}^\dagger \hat{a}_{\mathbf{k}} + \frac{1}{2\mathcal{V}} \sum_{\mathbf{k}\mathbf{k}'\mathbf{q}} V_{BB}(\mathbf{q}) \hat{a}_{\mathbf{k}'+\mathbf{q}}^\dagger \hat{a}_{\mathbf{k}-\mathbf{q}}^\dagger \hat{a}_{\mathbf{k}} \hat{a}_{\mathbf{k}'}. \tag{3}$$

Here $\mathcal{V}$ is the system volume, $\hat{d}_{\mathbf{p}}^\dagger$ and $\hat{a}_{\mathbf{p}}^\dagger$ are the creation operators of impurity and bosons, respectively, and $V_{BB}(\mathbf{q})$ and $V_{IB}(\mathbf{q})$ are the Fourier transforms of the respective interactions in Eq. (1). Next, the ladder operators of bosons are shifted using a simple canonical coherent state transformation leading to $\hat{a}_{\mathbf{p}}^\dagger \rightarrow \hat{a}_{\mathbf{p}}^\dagger + \delta_{\mathbf{p},0} \sqrt{N_0}$.

This is then followed by the crucial *Bogoliubov approximation*: All terms beyond quadratic order bosonic operators are neglected, leading to the truncated Hamiltonian

$$\hat{H}' = \sum_{\mathbf{p}} \frac{\mathbf{p}^2}{2m_I} \hat{d}_{\mathbf{p}}^\dagger \hat{d}_{\mathbf{p}} + \sum_{\mathbf{p}} \frac{\mathbf{p}^2}{2m_B} \hat{a}_{\mathbf{p}}^\dagger \hat{a}_{\mathbf{p}} + \frac{N_0}{\mathcal{V}} V_{IB}(\mathbf{0}) + \frac{\sqrt{N_0}}{\mathcal{V}} \sum_{\mathbf{k}'\mathbf{q}} V_{IB}(\mathbf{q}) \hat{d}_{\mathbf{k}'+\mathbf{q}}^\dagger \hat{d}_{\mathbf{k}'} (\hat{a}_{-\mathbf{q}}^\dagger + \hat{a}_{\mathbf{q}})$$

$$+ \frac{1}{\mathcal{V}} \sum_{\mathbf{k}\mathbf{k}'\mathbf{q}} V_{IB}(\mathbf{q}) \hat{d}_{\mathbf{k}'+\mathbf{q}}^\dagger \hat{d}_{\mathbf{k}'} \hat{a}_{\mathbf{k}-\mathbf{q}}^\dagger \hat{a}_{\mathbf{k}} + \frac{g_{BB} N_0^2}{2\mathcal{V}} + \frac{g_{BB} N_0}{2\mathcal{V}} \sum_{\mathbf{q} \neq 0} \left( 2\hat{a}_{\mathbf{q}}^\dagger \hat{a}_{\mathbf{q}} + \hat{a}_{\mathbf{q}}^\dagger \hat{a}_{-\mathbf{q}}^\dagger + \hat{a}_{\mathbf{q}} \hat{a}_{-\mathbf{q}} \right), \tag{4}$$

where, for simplicity, we have chosen the example of a boson contact interaction of strength $g_{\text{BB}}$. As a result of the Bogoliubov approximation the bosonic part of the model can be diagonalized using the standard Bogoliubov rotation

$$\hat{b}_{\mathbf{p}} = u_{\mathbf{p}}\hat{a}_{\mathbf{p}} + v_{-\mathbf{p}}^*\hat{a}_{-\mathbf{p}}^\dagger \,, \quad \hat{b}_{\mathbf{p}}^\dagger = u_{\mathbf{p}}^*\hat{a}_{\mathbf{p}}^\dagger + v_{-\mathbf{p}}\hat{a}_{-\mathbf{p}} \,. \tag{5}$$

While the Bogoliubov approximation thus allows to obtain a simple dispersion relation for bosonic quasiparticles, it, however, captures the effect of repulsion only within the Bogoliubov coefficients $u_{\mathbf{p}}$ and $v_{\mathbf{p}}$. They give rise, e.g., to the modified quasiparticle dispersion $\omega_{\mathbf{p}} = \sqrt{\epsilon_{\mathbf{p}}(\epsilon_{\mathbf{p}} + 2n_0 g_{BB})}$ where, notably, the *non-deformed*, homogenous boson density $n_0$ appears. It turns out that this approximate account of boson repulsion is insufficient to self-stabilize the Bose polaron. Indeed, the neglect of terms beyond quadratic order is responsible for the apparent instability of the truncated Hamiltonian (4)[2]. Instead, it is crucial to keep all terms beyond quadratic order which allows the repulsive interactions to act as a stabilizing counter term to the impurity attraction. Including these terms allows then to expand the theory not simply around the homogenous BEC but around a BEC that is already deformed due to the presence of the impurity (for a discussion of the one-dimensional case see Ref. [47]). As discussed in the following, this in turn allows one to effectively map the strong coupling Bose polaron problem onto a weakly interacting one.

## 2.1 Gross-Pitaevskii functional

Following this strategy we focus on a variational approach to the ground state of the full Hamiltonian (2). We use a product state [54]

$$\Psi(\mathbf{x}_1,\ldots,\mathbf{x}_N) = \phi(\mathbf{x}_1)\cdots\phi(\mathbf{x}_N), \tag{6}$$

where the condensate wave function $\phi(\mathbf{x})$ is normalized to the condensate particle number $\int d^d x \, |\phi(\mathbf{x})|^2 = N_0$[3]. Both $\phi(\mathbf{x})$ and the ground-state energy are found by minimizing the resulting variational Gross-Pitaevskii (GP) energy functional

$$E_{\text{GP}}[\phi] = \frac{\left[\mathbf{p}_0 - \int d^d x \, \bar{\phi}(-i\nabla)\phi\right]^2}{2m_{\text{I}}} + \int d^d x \left[\frac{|\nabla\phi|^2}{2m_{\text{red}}} + V_{\text{IB}}(\mathbf{x})|\phi(\mathbf{x})|^2 + \frac{g}{2}|\phi(\mathbf{x})|^4 - \mu|\phi(\mathbf{x})|^2\right]. \tag{7}$$

Here $\mu$ is the chemical potential and we assume weak boson repulsion represented in three dimensions by a contact interaction of strength $g = 4\pi a_{\text{BB}}/m_{\text{B}}$ with boson scattering length $a_{\text{BB}} > 0$. Note that normal ordering of the impurity kinetic term in Eq. (2) contributes to the boson kinetic term with reduced mass $m_{\text{red}}^{-1} = m_{\text{B}}^{-1} + m_{\text{I}}^{-1}$ [21,30,40].

The energy functional Eq. (7) exhibits two important limiting cases: (i) for an infinitely heavy impurity $m_{\text{I}} \to \infty$, the normal-ordered kinetic recoil term in the first line of (7) vanishes and the standard GP energy functional for bosons in a static external potential is recovered; (ii) for a Bose polaron at rest ($\mathbf{p}_0 = 0$), and a radially symmetric impurity potential $V_{\text{IB}}(|\mathbf{x}|)$, the wave function $\phi(\mathbf{x})$ is spherically symmetric and the recoil term again vanishes —it only re-appears beyond the product ansatz when boson correlations are included [40,55,56].

We find the condensate wave function $\phi(\mathbf{x})$ by minimizing the GP functional (7) in the thermodynamic limit subject to the boundary conditions $|\phi(\mathbf{x} \to 0)| < \infty$, $|\phi(|\mathbf{x}| \to \infty)| = \sqrt{n_0}$ in terms of the condensate density $n_0$ far away from the impurity. For a radially symmetric impurity potential $V_{\text{IB}}(|\mathbf{x}|)$ at rest ($\mathbf{p}_0 = 0$), the ground-state wave function is spherically symmetric and real. In the solution of the GP functional the energy is universally expressed in units of the BEC bulk chemical potential $\mu = gn_0$ and the distance from

---

[2]The instability becomes physical for a non-interacting BEC [30,38].

[3]For weak boson interactions $N \approx N_0$.

the impurity $r = |\mathbf{x}|$ is measured in units of the modified healing length $\xi = 1/\sqrt{2m_{\mathrm{red}}\mu}$ $= 1/\sqrt{8\pi(m_{\mathrm{red}}/m_{\mathrm{B}})a_{\mathrm{BB}}n_0}$, which involves the *reduced* mass and is therefore larger than the usual bulk healing length $\xi_0 = 1/\sqrt{2m_{\mathrm{B}}\mu}$ of the BEC without impurity.

We define the polaron energy functional $E[\phi] = E_{\mathrm{GP}}[\phi] - E_{\mathrm{GP}}[\phi_0]$ relative to the energy of the unperturbed BEC with wave function $\phi_0 = \sqrt{n_0}$. For the three-dimensional case it is conveniently expressed in terms of the scaled radial function $u(r) = r\phi(r)/\sqrt{n_0}$ as

$$\frac{E[u]}{\mu} = 4\pi n_0 \int_0^\infty dr \left[ \xi^2 \left( \left(\frac{du}{dr}\right)^2 - 1 \right) + \frac{V_{\mathrm{IB}}(r)}{\mu}u(r)^2 + \frac{(u(r)^2 - r^2)^2}{2r^2} \right]. \tag{8}$$

Here the boundary conditions for $\phi(\mathbf{x})$ translate to $u(0) = 0$ and $u(r \to \infty) = r$. Numerically, $E[u]$ is minimized by global optimization on $r \in [0, L]$ with $L = 10\xi$ and an $r$-grid spacing $\Delta r = 0.05\,\xi$ much smaller than the potential range.

We thus obtain the scaling solution of the GP equation (GPE) in units of $\mu$ and $\xi$ for a given dimensionless potential shape $\tilde{V}(x = r/\xi) = V_{\mathrm{IB}}(r)/\mu$. This scaling solution is *universal* for arbitrary values of the condensate density $n_0$, boson scattering length $a_{\mathrm{BB}}$ and mass ratio $m_{\mathrm{red}}/m_{\mathrm{B}}$ as long as GP theory applies [40, 42], with these parameters entering only indirectly via $\mu$ and $\xi$. For comparison with experiment, and in order to visualize the effect of boson repulsion, the universal solution can be rescaled to obtain the specific solution for desired values of the boson scattering length $a_{\mathrm{BB}}$ and the mass ratio $m_{\mathrm{red}}/m_{\mathrm{B}}$ in density units of energy $E_n = \hbar^2 n_0^{2/3}/2m_{\mathrm{red}}$ and length $n_0^{-1/3}$.

## 2.2 Impurity potential

For the impurity potential $V_{\mathrm{IB}}(r)$ we consider two different functional forms. This allows us to study the universality of the Bose polaron by analyzing how quasiparticle properties depend on the potential shape and range. Specifically, we consider an attractive Gaussian potential,

$$V_{\mathrm{gauss}}(r) = -V_0 \exp[-(r/R)^2], \tag{9}$$

and an exponentially decaying potential,

$$V_{\mathrm{expon}}(r) = -V_0 \exp[-r/R], \tag{10}$$

both of depth $V_0 > 0$ and range $R$.

The low-energy scattering properties of the impurity and boson are characterized by the impurity-boson scattering length $a_{\mathrm{IB}}$ and the effective range $r_{\mathrm{eff}}$, which determine the leading terms of the effective range expansion of the momentum-dependent scattering phase shift in three dimensions. They are found by numerically solving the Schrödinger equation for the scattering of a boson with the impurity in the center-of-mass system via the potential $V_{\mathrm{IB}}(r)$. Equivalently, one may solve the first-order nonlinear variable phase equation [57], which yields

$$a'(r) = 2m_{\mathrm{red}}V_{\mathrm{IB}}(r)[r - a(r)]^2,$$
$$r'_{\mathrm{eff}}(r) = -4m_{\mathrm{red}}V_{\mathrm{IB}}(r)r^2\left(\frac{r}{a(r)} - 1\right)\left(\frac{r}{3a(r)} - 1 + \frac{r_{\mathrm{eff}}(r)}{r}\right). \tag{11}$$

Here $a(r)$ and $r_{\mathrm{eff}}(r)$ obey the boundary conditions $a(0) = 0$, $r_{\mathrm{eff}}(0) = 0$, and account for the phase shift accumulated by the scattering wave function (for the generalization of the variable phase equation to singular potentials see [58]). Correspondingly, the differential equations (11) are integrated from $r = 0 \ldots \infty$ which yields the scattering length $a_{\mathrm{IB}} = a(r \to \infty)$ and effective range $r_{\mathrm{eff}} = r_{\mathrm{eff}}(r \to \infty)$.

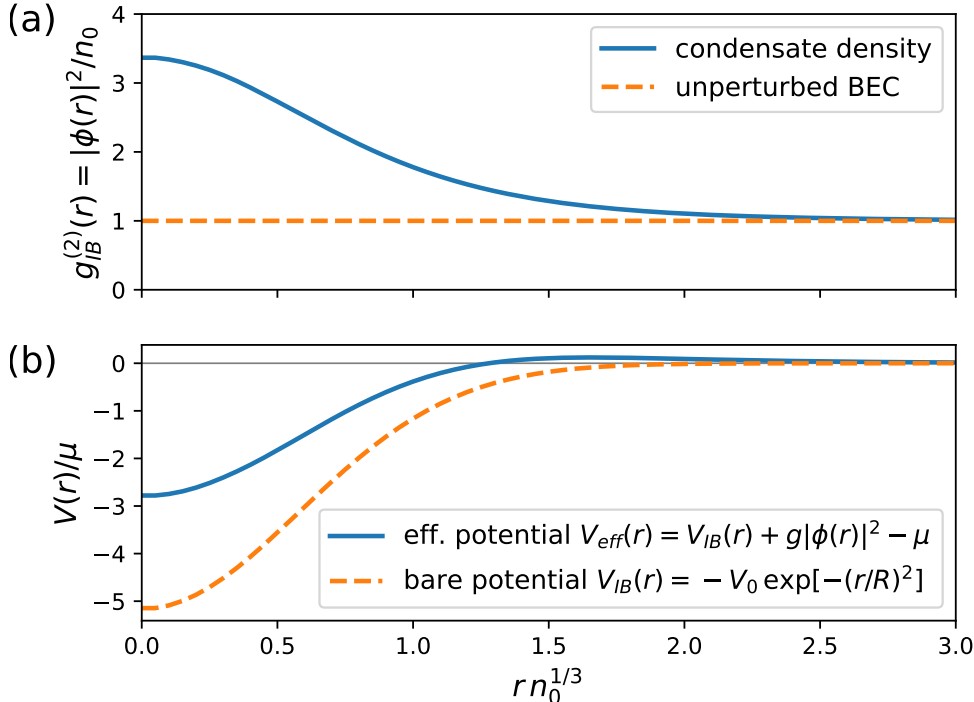

Figure 2: (a) Impurity-boson density-density correlation function $g^{(2)}_{\mathrm{IB}}(r)$ as determined from the condensate wave function $\phi(r)$ as function of the distance $r$ from the impurity (blue solid). For an attractive impurity potential the wave function is enhanced near the impurity as compared to that of an unperturbed BEC $\phi_0(r) = \sqrt{n_0}$ (orange dashed). (b) The bare impurity potential (orange dashed) of Gaussian shape ($V_0/\mu = 5.1651$, $R/\xi = 0.81892$) has an effective range $r_{\mathrm{eff}} = \xi$ and positive scattering length $a_{\mathrm{IB}} = 4\xi$, and correspondingly admits a bound state. An extra bosonic test particle is, however, subject to the effective potential (blue solid) that is weakened by the repulsion from the polaron cloud; while still attractive the effective potential is characterized by an effective, negative scattering length $a_{\mathrm{IB,eff}} = -0.1\xi$ (and renormalized $\tilde{r}_{\mathrm{eff}} = -207\xi$) which thus no longer supports a bound state. Hence, additional bosonic quantum fluctuations lead only to a weak, additional dressing of the impurity particle.

## 3 Effective potential

First, we present results for the condensate profile around the impurity that we find by minimizing the Gross-Pitaevskii energy functional (7)–(8). For attractive impurity-boson interaction, the wave function $\phi(r)$ —which, in the comoving frame, directly yields the impurity-boson density-density correlation function $g^{(2)}_{\mathrm{IB}}(r) = |\phi(r)|^2/n_0$— is enhanced near the impurity, as shown in Fig. 2(a). In this figure we have chosen a potential $V_{\mathrm{IB}}(r)$ (dashed line in Fig. 2(b)) that is sufficiently deep to support a two-body bound state at energy $-\varepsilon_B < 0$. Correspondingly, the potential is characterized by a positive impurity-boson scattering length, in Fig. 2(b) chosen as $a_{\mathrm{IB}} \approx 4\xi > 0$.

We thus realize a scenario as described in the introduction, which is mimicked by a microtrap or a single-site Bose-Hubbard model. While at the two-body level it suggests a dynamical instability where the occupation of the bound state would grow without bounds, many-body effects come to the rescue. Indeed, due to the repulsion between bosons, each additional boson

trying to participate in the formation of the Bose polaron is subject to the effective potential

$$V_{\text{eff}}(r) = V_{\text{IB}}(r) + g\,|\phi(r)|^2 - \mu, \tag{12}$$

that results from by the competition of the bare attractive potential $V_{\text{IB}}(r)$ (dashed line in Fig. 2(b)) and the repulsion created by the already existing polaron cloud.

In essence, this effect can be understood as arising in an effective density-functional theory (DFT) for the BEC particles in the presence of the impurity: the bosons already attracted to the impurity screen the attractive potential and make it shallower, as shown by the blue line in Fig. 2(b). Moreover, we find that they create a small repulsive barrier at intermediate distance. As a consequence, the *effective* potential no longer admits a bound state and it is correspondingly characterized by a *negative* effective impurity-boson scattering length $a_{\text{IB,eff}} \approx -0.1\xi$. Hence, the single-particle excitation spectrum for each additional boson is bounded from below: the dynamical instability is replaced by *Bose polarons self-stabilized by their dressing cloud*.

## 4 Bose polaron energy and contact

The value of the Gross-Pitaevskii functional (7) evaluated at the ground-state wave function determines the polaron energy $E = E[\phi]$ relative to the homogeneous BEC. The energy is shown in Fig. 3(a) as a function of the dimensionless impurity-boson interaction $\xi/a_{\text{IB}}$. It is always negative for an attractive impurity potential. In Fig. 3(b) we present the energy for a mobile impurity of arbitrary mass in units of $E_n$ which depends on the BEC density $n_0$. Specifically, we show results for two BEC gas parameters, which for equal mass of boson and impurity, i.e., $m_{\text{red}}/m_{\text{B}} = 1/2$, correspond to values $n_0 a_{\text{BB}}^3 = (4\pi)^{-3} = 5.0 \times 10^{-4}$ ($n_0\xi^3 = 1$) at stronger boson repulsion and $n_0 a_{\text{BB}}^3 = (8\pi)^{-3} = 0.63 \times 10^{-4}$ ($n_0\xi^3 = 2.8$) at weaker boson repulsion. We find that, in absolute terms, the polaron binding energy is larger for weaker boson repulsion where the BEC can be more strongly deformed and thus acquire more attractive potential energy.

At weak impurity-bath attraction $1/a_{\text{IB}} n_0^{1/3} \ll -1$ the polaron energy approaches the mean-field value

$$E_{\text{mf}} = \frac{2\pi a_{\text{IB}} n_0}{m_{\text{red}}} = 4\pi a_{\text{IB}} n_0^{1/3} E_n, \tag{13}$$

irrespective of $a_{\text{BB}}$. Within mean-field theory the polaron energy diverges to $-\infty$ at unitarity $1/a_{\text{IB}} = 0$. Variational approaches based on the *truncated* Hamiltonian Eq. (4) predict that the inclusion of Bogoliubov corrections is not sufficient to heal this instability, but rather results in a shift of the instability to the repulsive side of the Feshbach resonance [30]. In contrast, we find that going beyond the Bogoliubov approximation by working with the full Hamiltonian (1) allows the boson repulsion to stabilize the polaron at a finite ground-state energy that smoothly crosses over from the attractive to the repulsive side of the Feshbach resonance.

The deformation of the BEC is also reflected in the number of bosons participating in the formation of the polaron dressing cloud

$$N_{\text{cloud}} = 4\pi n_0 \int_0^\infty dr[u(r)^2 - r^2]. \tag{14}$$

As shown in Fig. 3(c), for a smaller gas parameter (orange dashed line) the impurity attracts a larger polaron cloud because the bosons are less repulsive. Naturally, this larger dressing cloud corresponds to the larger polaron binding energy found in Fig. 3(b).

Our results in Fig. 3 are shown for a constant range $R$ across the Feshbach resonance as applicable to experiments where the microscopic range of interactions can typically not

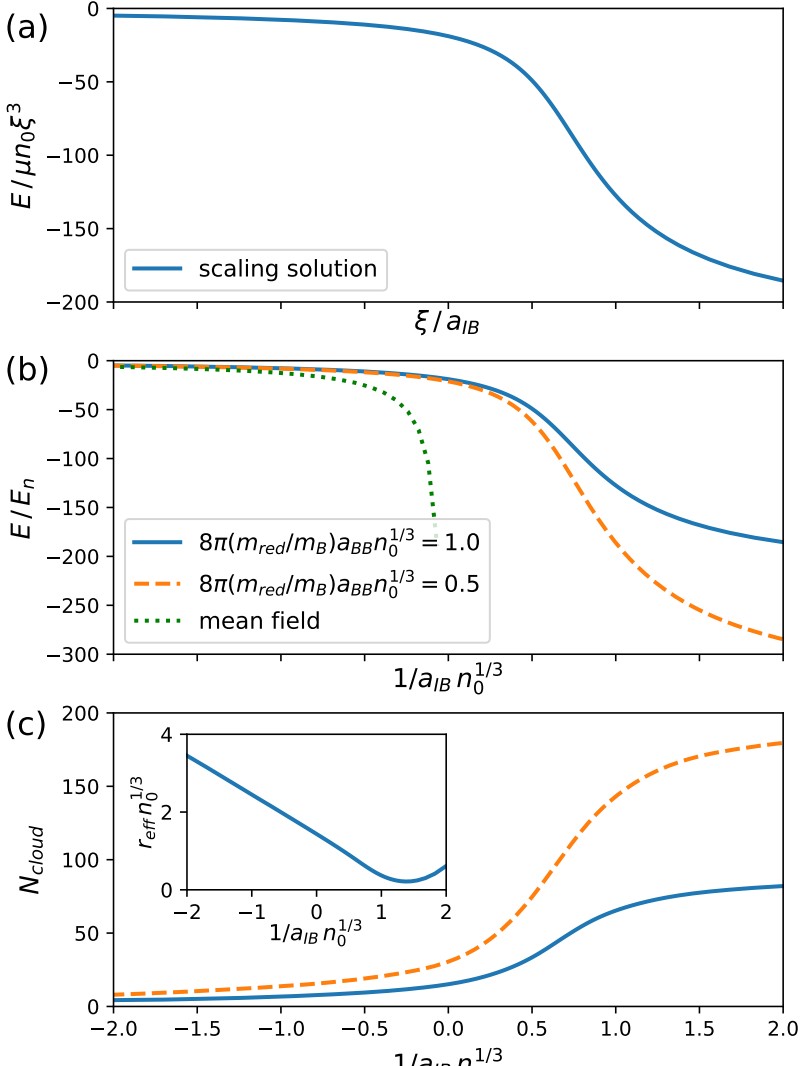

Figure 3: Bose polaron energy across an impurity-boson Feshbach resonance. (a) Scaling solution for the polaron energy $E/\mu n_0 \xi^3$ as function of the impurity-Bose interaction $\xi/a_{\mathrm{IB}}$ for a Gaussian potential of fixed range $R = \xi$. (b) Polaron energy $E$ in density units $E_n = \hbar^2 n_0^{2/3}/2m_{\mathrm{red}}$ in dependence on the impurity-Bose interaction $1/a_{\mathrm{IB}} n_0^{1/3}$ for different BEC gas parameters. The polaron binding energy is larger for weak boson repulsion $8\pi(m_{\mathrm{red}}/m_{\mathrm{B}})a_{\mathrm{BB}} n_0^{1/3} = 0.5$ (orange dashed) compared to stronger repulsion $8\pi(m_{\mathrm{red}}/m_{\mathrm{B}})a_{\mathrm{BB}} n_0^{1/3} = 1.0$ (blue solid); at weak coupling it approaches the mean-field result (13) (green dotted). (c) Particle number $N_{\mathrm{cloud}}$ within the polaron dressing cloud. Inset: the effective range $r_{\mathrm{eff}}$ for fixed potential range $R n_0^{1/3} = 1$ is smallest on the repulsive side.

be tuned synchronously with the scattering length. Thus the effective range $r_{\mathrm{eff}}$ varies in dependence on $a_{\mathrm{IB}}$: The inset of Fig. 3(c) shows the effective range that is obtained from the scattering phase shift using Eq. (11). At constant potential range $R n_0^{1/3} = 1$, the effective range $r_{\mathrm{eff}}$ reaches a minimum on the repulsive side of the resonance ($a_{\mathrm{IB}} > 0$) and grows towards both weak-coupling limits $a_{\mathrm{IB}} \to 0^{\pm}$.

The variation of the polaron energy with $a_{\mathrm{IB}}$ defines the impurity contact parameter [38, 40, 42, 51, 59] which characterizes the impurity-boson correlations $g_{\mathrm{IB}}^{(2)}(r)$ at short distances

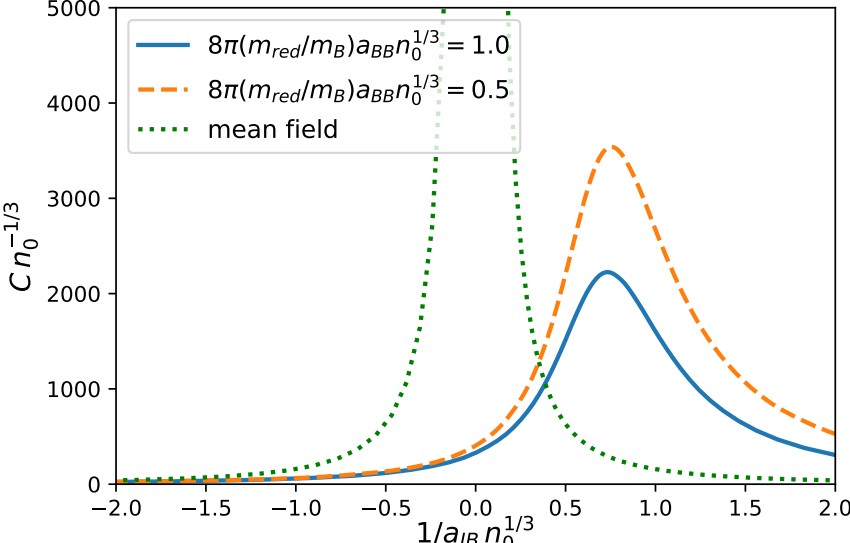

Figure 4: Tan contact $C$ of the Bose polaron for different boson repulsion (BEC gas parameter) across the impurity-boson Feshbach resonance. The contact $Cn_0^{-1/3}$ obtained from Eq. (15) reaches its maximum on the repulsive side of the resonance and increases for weaker boson repulsion. At weak coupling it approaches the mean-field result (16) (green dotted).

outside the impurity potential:

$$C = \frac{8\pi m_{\text{red}}}{\hbar^2} \frac{\partial E}{\partial(-1/a_{\text{IB}})} = 4\pi n_0^{1/3} \frac{\partial(E/E_n)}{\partial(-1/a_{\text{IB}} n_0^{1/3})} \,. \tag{15}$$

The contact is shown in Fig. 4: we find that it reaches a maximal value on the repulsive side of the resonance. Similar to the energy, we find that the contact is larger for smaller boson repulsion $a_{\text{BB}}$ (blue solid line) where the BEC is more strongly deformed and thus $g_{\text{IB}}^{(2)}(r)$ is enhanced (see Fig. 2). At weak attractive interaction the contact approaches the ground-state value of an impurity in an ideal BEC ($a_{\text{BB}} = 0$) [38]

$$C_{\text{mf}} = 16\pi^2 n_0 a_{\text{IB}}^2 \,, \tag{16}$$

which agrees with the mean-field result prediction.

## 5 Universality

Finally, we test the notion of universality of the Bose polaron by studying different shapes and ranges $R$ of the impurity potential. Generally, we work in the regime where the potential range is much larger than the boson scattering length, $R \gg a_{\text{BB}}$, where local Gross-Pitaevskii theory has been shown to be applicable [35]. To test universality in this regime, we specifically compare the predictions following from the Gaussian potential $V_{\text{gauss}}(r)$ in Eq. (9), and the exponential potential $V_{\text{expon}}(r)$ in Eq. (10) for various ranges $R$.

We find that when tuning the depth of the potentials to obtain equal scattering length $a_{\text{IB}} = -\infty$ at equal range $R$, the polaron energies are very different. Instead, if $R$ is tuned to yield the same effective range $r_{\text{eff}}$ for both potentials, as shown in Fig. 5(a), remarkable agreement between the polaron energies is found. Indeed, as shown in Fig. 5(b), for both

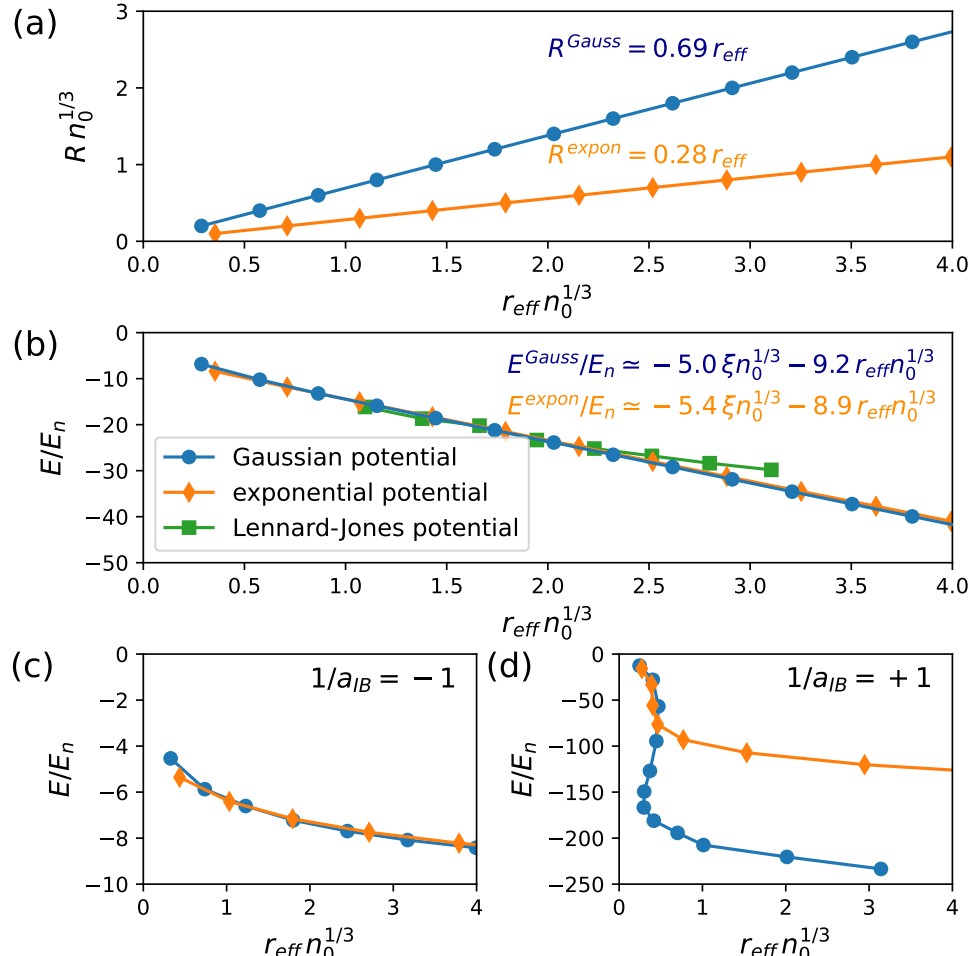

Figure 5: Universality of the Bose polaron. (a) In order to yield the same effective range $r_{\text{eff}}$, the Gaussian and exponential potentials need to be tuned to different potential ranges $R$; here shown for fixed unitary scattering length $1/a_{\text{IB}} = 0$. (b) Polaron energy $E$ as function of the effective range $r_{\text{eff}}$ at unitarity $1/a_{\text{IB}} = 0$ for two different potential shapes. The boson repulsion is set to $8\pi(m_{\text{red}}/m_{\text{B}})a_{\text{BB}}n_0^{1/3} = 1$. The polaron energy coincides for both potential shapes and increases linearly with $r_{\text{eff}}$. This universality also extends to Lennard-Jones potentials as shown by the green squares. (c) The polaron energy at negative scattering length $1/a_{\text{IB}}n_0^{1/3} = -1$ is universal for $r_{\text{eff}}n_0^{1/3} \gtrsim 1$ for the parameters chosen. (d) Polaron energy at positive scattering length $1/a_{\text{IB}}n_0^{1/3} = 1$. For the Gaussian potential there is no unique polaron energy for intermediate values of $r_{\text{eff}}$ (see text).

potentials, the polaron energy approximately follows a *linear* scaling law for $r_{\text{eff}} \gtrsim 0.2\xi$,

$$\frac{E(1/a_{\text{IB}} = 0)}{E_n} = -5.2(2)\,\xi n_0^{1/3} - 9.0(1)\,r_{\text{eff}}n_0^{1/3}. \tag{17}$$

Remarkably, this universality is found not only for the purely attractive Gaussian and exponential potentials but extends also to interactions featuring a repulsive contribution such as a Lennard-Jones potential $V_{\text{LJ}}(r) \sim \lambda R^{10}/r^{12} - R^4/r^6$, as illustrated in Fig. 5(b). This shows that in the Bose polaron problem the momentum-dependent scattering phase shift is probed in the regime where the effective range expansion is valid. Our result complements a recent GPE study which found a power-law scaling of the unitary polaron energy at ranges shorter

than the healing length, $E/E_n \sim (r_{\text{eff}}/\xi)^{1/3}$ for $a_{\text{BB}} \lesssim r_{\text{eff}} \ll \xi$ in the case of a square-well potential [42]. The GPE approach can be extended to even shorter ranges $r_{\text{eff}} \ll a_{\text{BB}}$ by using a nonlocal generalization of Gross-Pitaevskii theory [40].

Also on the attractive side of the resonance, see Fig. 5(c) for $1/a_{\text{IB}}n_0^{1/3} = -1$, we find a universal effective range dependence of the polaron energy, albeit not a linear one. On the repulsive side, instead, to cover the domain of effective ranges shown in Fig. 5(d), for our parameters the Gaussian potential, unlike the exponential, must be tuned over such depths that it supports more than one bound state. Consequently, there is no longer a unique mapping from $r_{\text{eff}}$ to $E$. This is illustrated in Fig. 5(d) for the repulsive side, at $1/a_{\text{IB}}n_0^{1/3} = 1$: universality can at best hold in the vicinity of the first bound state, and only for a limited interval of $r_{\text{eff}}$ values. Furthermore, since the Efimov effect modifies the polaron energy spectrum [2,3,49], universality can also depend on the three-body parameter [50,60–62].

## 6  Discussion

The variational principle provides a powerful tool to compute both ground-state and dynamical properties of quantum many-body systems. However, in order to make full use of its predictive power it is essential to understand the limitations of approximations applied to the Hamiltonian to be analyzed. In this regard the strong-coupling Bose polaron is a case in point. The full Bose polaron Hamiltonian (1) —and equivalently Eq. (3)— is bounded from below for repulsive boson interaction, and hence variational wave functions give rigorous bounds on the ground-state energy. For finite Bose repulsion, the polaron energy remains finite for any Bose-impurity scattering length, including resonant interactions, and the ground state represents a strong-coupling Bose polaron that is self-stabilized by its own dressing cloud.

In contrast, when applying the Bogoliubov approximation to the full model (1) by truncating terms of higher-than quadratic order in the boson operators, the resulting, truncated Hamiltonian $\hat{H}'$ in Eq. (4) becomes unbound from below. Crucially, this results in an instability of the Bose polaron problem that is solely an artefact of this approximation.

Quite remarkably, the instability of the truncated Hamiltonian $\hat{H}'$ becomes, however, only evident when considering wavefunctions that account for more than two phonon excitations from the homogenous BEC. For instance, for a Chevy-type wavefunction [25,27,28] that itself is truncated at the single excitation level, the terms beyond quadratic order in the repulsive interactions have vanishing expectation value. Thus, incidentally, the Chevy ansatz yields *the same* prediction when applied to both the full and the truncated model. Thus due to its tremendous simplicity the Chevy ansatz becomes immune to the instability of approximate Hamiltonian $\hat{H}'$.

However, while being a well-defined approach, the simple Chevy ansatz misses the fact that at the weak Boson repulsion (as typically present in cold gases) the polaron cloud—even within the full model Eq. (1)—can contain an exceedingly large number of bosons (Fig. 3(b)). Such a large local deformation of the BEC is naturally captured by the inhomogeneous wave function (6). Crucially, while accounting for an arbitrary number of boson excitations, when applied to the full Hamiltonian (1) it still leads to a bounded energy functional (7). Its solution shows that the smaller the boson repulsion and the wider the impurity potential, the larger the polaron cloud becomes. The product state approach is complementary to the Chevy ansatz including its extensions to multi-boson excitations [3,49,63], and it becomes particularly accurate for soft impurity potentials where it is justified to ignore bosonic correlations. Remarkably, the case of extremely soft potentials is realized with Rydberg excitation immersed in BECs. In this case it was predicted that up to hundreds of atoms can be bound to the single impurity leading to the creation of Rydberg polarons [64]. Since for Rydberg impurities the range of interactions

*R* dramatically exceeds the interparticle distance, our local GP theory applies and provides a so far missing explanation as to why the experimental observation of Rydberg polarons [29] is described exceptionally well by a coherent state approach [26].

Recently also first steps to the understanding of the complementary, intermediate regime of short-range impurity potentials —with yet large dressing clouds— has been achieved by using an extension to nonlocal Gross-Pitaevskii theory [40]. In conjunction with our present result, these combined new approaches resolve a fundamental shortcoming of the Bogoliubov approximation: while the formulation of the interacting Bose gas in terms of Bogoliubov quasi-particles is exact, the additional approximation to neglect the residual interaction between phonons is not. In particular, the quadratic Bogoliubov mean-field Hamiltonian is unbounded from below for the strong-coupling Bose polaron, most obviously in the regime where a two-body bound state appears on the repulsive side of the resonance [30, 31].

As discussed above, the Chevy-type ansatz applied to the truncated Bogoliubov Hamiltonian [28] still yields a finite polaron energy since it is of such low order in boson excitations that it is not sensitive to the truncated part of the Hamiltonian. However, when the coherent state ansatz or higher-order excitation extensions of the Chevy ansatz are applied to the truncated Hamiltonian (4) they can lead to divergencies in the ground-state energy. In the case of the coherent state ansatz [30, 36] the divergence is due to the large occupation of excitations in the vicinity of the impurity not counteracted by boson repulsion. Instead the local extension to the truncated Bogoliubov approach studied in the present work (see also [42, 50, 51]) as well as its nonlocal counterpart [40] provides a stable starting point for strong-coupling Bose polaron dynamics, and we showed how it can find an effective description in terms of a renormalization of the impurity-boson potential (Fig. 2).

Beyond our treatment of the two-particle impurity-boson correlations, three-body and higher-order correlations give rise to the Efimov effect and three-body recombination. The Efimov effect can occur either between one impurity and two bosons [3, 49, 63] or between two impurities and one boson [2, 6]. These few-body effects can be captured by Gaussian variational wave functions [55, 56]. Alternatively, extensions of the Chevy ansatz to two or more independent bosonic excitations [3] can be employed. The latter approach was applied in the analysis of the truncated model (4) and universal scaling depending on the three-body parameter was found [49, 63]. It remains an interesting open question how universal three-body physics carries over to the many-body case in a dense bosonic medium when the full Hamiltonian Eq. (1) is considered [65].

Finally, in ultracold atomic gases the boson repulsion originates from an attractive van-der-Waals potential between atoms. This results in the existence of deeply bound molecular states into which atoms can decay in three-body recombination. These deeply bound states are neither accounted for in variational approaches nor in quantum Monte Carlo [43]. Up to now most experiments probe the metastable Bose polaron state of matter on transient time scales where the deeply bound states —arising both from the fundamentally attractive Bose-Bose and Bose-impurity potentials— can be ignored, and thus the stable variational approach discussed in our work is well applicable. However, as one starts to explore longer time scales or the build-up of more complex correlated states of impurities, understanding the impact of the fundamental dissipative nature arising from deeply bound states becomes essential and requires the development of new theoretical approaches to quantum impurity problems.

## Acknowledgments

This work is supported by the Deutsche Forschungsgemeinschaft (DFG, German Research Foundation), project-ID 273811115 (SFB1225 ISOQUANT) and under Germany's Excellence

Strategy EXC2181/1-390900948 (the Heidelberg STRUCTURES Excellence Cluster). R. S. is supported by the Deutsche Forschungsgemeinschaft (DFG, German Research Foundation) under Germany's Excellence Strategy – EXC-2111 – 390814868 (Excellence Cluster 'Munich Center for Quantum Science and Technology').

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
