# Peer review of "Self-stabilized Bose polarons"

_SciPost Physics, doi:SciPost Phys. 13, 054 (2022)_

## Round 1 · Referee Report · Anonymous (Referee 1) · 2021-3-31

Strengths

  1. Provides a simple picture of stabilization of the Bose polaron with a simple theory, whereas previous theories fail to account for this stabilization.

Weaknesses

  1. The range of validity of the theory is not sufficiently discussed.

Report

This is an interesting work investigating the stabilization of the Bose polaron by the repulsion between bosons, in the framework of the Gross-Pitaevskii theory. It constitutes an important contribution since this stabilization was absent in most theories. The paper is very clearly written and the results are physically sound. All of the journal's acceptance criteria are met, and I think it duly deserves publication once the authors have clarified a few points and considered the following suggestions.

Requested changes

Here are some points I would like to clarify:

1) The authors should specify more clearly the range of validity of their theory. In Section III.5, they state "our local Gross-Pitaevskii theory is reliable as long as the potential range $R$ is not so short that the assumption of a slowly varying potential is violated". With respect to which length scale should the range R not be too short? Later they hint that the theory could be extended to $r_{eff} \ll a_{BB}$, so I suppose the relevant length is $a_{BB}$. If so, why is it the relevant length scale? This should be discussed more explicitly.

2) The authors investigate the universality of their results by considering two different potentials: a Gaussian and an exponential potential. These two potentials are relatively similar. To get a more convincing evidence of universality, it would be interesting to consider a markedly different potential. For instance, if it is not too much work for the authors, it would be interesting (and more physically motivated) to use a van der Waals potential, such as a Lennard-Jones potential. This is just a suggestion.

3) In Fig. 5, it would be instructive to plot the impurity-boson dimer energy for comparison. If possible, it would also be interesting to plot results for vanishing n0. Intuitively, that limit should brige the mean-field regime and the dimer energy.

4) In Fig. 5, it is stated that the polaron energy at negative scattering length is universal for $r_{eff} \gtrsim \xi$. Is it a numerical observation for that particular calculation or a more general statement? The authors should clarify the generality of this statement.

5) It is pointed out that a deep Gaussian potential can have several bound states, which makes the polaron energy non-unique. At first, it is not clear why an emphasis is put on the Gaussian potential. It would seem that the exponential potential would equally have several bound states if sufficiently deep. I presume that in the range of effective ranges considered, the required depth of the potentials is such that a second bound state appears for the Gaussian potential, but not for the exponential potential. Is that right? Somehow this discussion could be clarified. It might help to specify either in the text or on the curves of Fig 5(d) the number of bound states supported by each potential.

I have a few additional comments:

6) The authors explain that the stabilization of the Bose polaron comes from the repulsion between the bosons. However, it has been shown in Ref. [55] that even for non-interacting bosons, the interaction between the impurity and bosons can have a stabilizing effect. The authors may want to comment on this additional effect and whether it is present in their theory (presumably not) and what it would take to include this effect.

7) In section II.A, the authors state that their ansatz "is based on a coherent ansatz". I think the product form they consider is actually a Fock state, and not a coherent state.

8) In Section VI, the authors state "However, when the coherent state ansatz or higher-order excitation extensions of the Chevy ansatz [41, 55] are applied to the truncated Hamiltonian (4) they lead to the aforementioned, spurious divergence of the ground-state energy". I do not think that there is any spurious divergence of the ground-state energy in Refs. [41][55]. Perhaps the authors meant to refer to Ref. [30]?

---

## Round 1 · Referee Report · Anonymous (Referee 2) · 2021-4-21

Strengths

  • A relatively simple solution to the Bose polaron problem that removes the divergence in the energy.

Weaknesses

  • The GP theory is unlikely to be valid for short-range impurity-boson potentials. This should be more clearly discussed.

Report

This paper certainly meets the requirements for publication. It investigates the behavior of an impurity in a BEC, a topic of intense interest in quantum gases, and it provides a way to remove the divergence in the ground-state energy that has plagued many previous approaches, such as Ref [30]. It also presents a simple classical picture of how bosons effectively screen the impurity and thus reduce the strength of the attractive impurity potential.

The main points that require more discussion are the validity of the classical-field approach and how the current work fits with other theoretical approaches. Specifically, I would like the authors to address the following questions/comments:

  1. What is the precise condition for the classical field (GP) approach to be valid? I would expect that the range $R$ has to be sufficiently large but does it need to be larger that the healing length or just $a_{BB}$?

  2. How does that the current approach differ from that in Ref. [38]? The authors of Ref [38] appear to use the same GP equation, yet claim a larger regime of validity, i.e., $R \sim a_{BB}$.

  3. Does the approach in the paper capture the perturbative results in the limit of weak boson-impurity coupling? For instance, it is known that the GP approach in Ref [43] does not recover it properly, while the coherent-state approach in Ref. [30] does.

  4. I was confused about why it was claimed that higher order extensions of the Chevy ansatz lead to a spurious divergence. In Ref. [41] the dependence on a high-momentum cutoff is due to Efimov physics, which is a physical effect that is also present in QMC calculations.

  5. Can the approach be extended to non-zero temperature?

Requested changes

  1. A clear discussion of the validity of the GP theory in the paper.

  2. More comparison with previous work. In particular, how does the theory compare with the QMC calculations for an ionic impurity in Ref. [44]?

  3. Can the authors comment on the limit of a zero-range potential? Since this has been used in other works including QMC calculations.

  4. The authors should also comment more on Ref [55], which shows that an infinitely heavy impurity can trap an infinite number of bosons but still have a lower bound on the ground-state energy. An infinite number of bosons trapped by a short-range potential was also shown in Chen et al., PRA 98, 041602(R) (2018).

---

## Round 1 · Referee Report · Anonymous (Referee 3) · 2021-5-4

# Report Scipost "Self-stabilized Bose polarons" by Richard Schmidt and Tilman Enss.

In the present work the authors elaborate on the stabilization mechanism of Bose polarons occurring for strong local attractions. Indeed, it is known that using conventional approaches, e.g. the Fröhlich model, an instability emerges in this strongly attractive interaction regime and the bosons of the medium accumulate towards the impurity. To tackle this problem a variational type approach is constructed that operates beyond the standard Bogoliubov approximation including the local boson repulsion which acts against the impurity's attractive potential excluding binding. As argued this additive term is able to stabilize the Bose polaron in the vicinity of the respective scattering resonance. For instance, it is demonstrated that the polaron energy is bounded from below across the resonance while the polaron dressing cloud remains sizable. Moreover, at resonance, the polaron energy shows a universal behavior on the effective range and the impurity contact is finite exhibiting a peaked structure at positive scattering lengths.

I find the results and ideas presented intriguing with direct experimental relevance and most likely will inspire similar theoretical investigations in the future. The manuscript is also well written. However, I have some questions regarding the findings and also the applicability of the used method. Thus, if the authors provide convincing answers to my comments, summarized below, and perform the respective minor revisions then I would certainly recommend this work for publication in Sci post.

1) I have some conceptual questions regarding the applicability of the used model, e.g. described by Eqs. (1) or (2).

a) As far as I understand it holds for both mobile and immobile impurities (Figure 3(b)). However, I am not able to clearly judge its applicability for finite impurity masses, meaning that in the latter case impurity-medium correlations is possible to become important. Can the authors briefly comment on this issue?

b) Is it possible the present formalism to be extended for larger impurity concentrations, e.g. more than a single one? Would then its reliability be restricted?

c) Can it be extended to account also for long-range interactions e.g. dipoles?

I imagine that the emergent physics especially in c) will be fundamentally different than the one described in the manuscript but I wonder what is the authors opinion here.

2) On page 3, first paragraph of the left column, it is stated that "For bosons in the vicinity … leads to induced interactions between bosonic particles ...". What type of induced interactions are meant here? Also, are they always attractive or is it possible to engineer also repulsive ones?

3) Is it possible to compare or translate the modified quasiparticle dispersion to the one obtained for particle-balanced mixtures e.g. introduced in Phys. Rev. Lett. **117**, 100401 (2016) by Petrov? I can understand that this might be well beyond the scope of the present work but

if the authors are able to provide any relevant hint would be extremely useful not only for advancing the impact of their work but also for the community.

4) On page 3, first paragraph of the right column, I would suggest besides Ref. [40] to include also Refs. New J. Phys. **21**, 103026 (2019), New J. Phys. **22**, 043007 (2020), that also refer to the one-dimensional case and elaborate on the effect of attractive impurity-boson interactions while relying on a variational treatment.

5) It is also not clear to me how the present model accounts for impurity-medium correlations. Please comment. I would expect that such effects would become particularly important in the nonequilibrium polaron dynamics which the authors claim to be a next step to apply their formalism.

6) Regarding the dynamical instability appearing in the GPE. Is it related to the presence of Bogoliubov modes with complex eigenfrequencies leading to the instability of the corresponding stationary solution of the GPE solution? Or is it one of the so-called thermodynamic ones as introduced e.g. by Castin in arXiv:cond-mat/0105058?

7) I wonder what is the behavior of the polaron residue in the interaction region of the bound state? Is it finite? If no, how then the polaron picture should be interpreted?

8) I am bit preplexed regarding the results shown in figure 3 (b). On the one hand in the caption it is argued that the polaron binding energy is presented when equal mass of boson and impurity is considered and for two different gas parameters. However, in the text it is written that "...we present the energy for a mobile impurity of arbitrary mass...". Please comment. Also, why the authors do not show the polaron binding energy for a heavier impurity and one of the scattering lengths used for the equal mass case? This will enable to judge also the the impact of the mass on the binding energy.

---

## Round 2 · Referee Report · Anonymous (Referee 1) · 2021-8-13

Strengths

  1. Provides a simple picture of stabilization of the Bose polaron with a simple theory, whereas previous theories fail to account for this stabilization.

Weaknesses

Addressed.

Report

Although the changes to the manuscript are minimal, I find them acceptable.

In particular, I thank the authors for clarifying the validity of their theory, as requested by other refereees, which was the most important point.

Requested changes

There are two points that I would like the authors to seriously consider before they publish their work.

  1. In their response, the authors argue that atomic interactions typically support a large number of bound states, which makes it too difficult to use a van der Waals potential in their work. The authors could have nonetheless considered a shallow van der Waals potential with only one bound state (such as that of helium) to compare with the Gaussian/Exponential potential and check their claimed universality. This would not be particularly more difficult than a Gaussian or exponential potential with one bound state, and would significantly strengthen the manuscript. If the authors do not do this simple check in the present work, it will need to be be done in another publication (if it is ever done). I think this is a good opportunity for the authors, but again I leave it to their discretion.

  2. I find it regrettable that the authors insist on referring to their ansatz as a "coherent state". Technically, as I pointed out in my first report, they introduce it as a Fock state, not as a coherent state, and it is equivalent with a coherent state only in the limit of large numbers of particles. Moreover, the coherent nature (superposition of states with different numbers of particles) does not play any role in this work. I think terms like "Fock state" or "mean-field" or "Gross-Pitaevskii" would be much more easily understood. Moreover, they oppose their work to the coherent ansatz of Ref. [30,36], which truly is a coherent ansatz where the superposition of number states plays a crucial role (here, the number of excitations). Naming both approaches as coherent ansatz may cause confusion to the readers.

As an aside, concerning the divergence of the ground-state energy found in some models, I am satisfied with the changes in the manuscript, which now cites the correct references where this divergence occurs and its plausible cause ("large occupation of excitations in the vicinity of the impurity not counteracted by boson repulsion"). But surprisingly, in their response, the authors give another explanation related to the Thomas collapse for zero-range interactions. This has nothing to do with the explanation given in the manuscript, and I highly doubt that this is correct. The Thomas collapse, as noted by the authors, occurs at any fixed value of the scattering length a, whereas the divergence found in these models occurs for 1/a larger than a critical value. Again, I am satisfied with what is written in the manuscript, but I just wanted to clarify this point in case there was some confusion.

---

## Round 2 · Referee Report · Anonymous (Referee 2) · 2021-9-3

Report

The authors have gone some way towards addressing the comments of the referees, particularly regarding the regime of validity of their theory. However, there are still some remaining points that I would like to see addressed before publication:

  • If the regime of validity corresponds to $R\gg a_{BB}$, then presumably this is why the results of the GP theory do not agree with the QMC results in [Phys. Rev. Lett. 127, 033401 (2021)] ? This should be mentioned in the manuscript.

  • I do not think that the perturbative limit of weak boson-impurity coupling violates the assumption of $R\gg a_{BB}$, as claimed by the authors. Rather, one requires $a_{IB}$ to be small compared to the BEC healing length, and this can be satisfied by taking the boson density to be sufficiently low, without making any assumptions about the range of the potential. Therefore, if the GP theory does not match the perturbative weak-coupling results, this suggests that the theory cannot accurately capture the low-density limit. The authors should at least comment on this in the manuscript.

Requested changes

See above

---

## Round 2 · Author Response

Reply to Report 1

Referee: Strengths: Provides a simple picture of stabilization of the Bose polaron with a simple theory, whereas previous theories fail to account for this stabilization. Weaknesses: The range of validity of the theory is not sufficiently discussed.

This is an interesting work investigating the stabilization of the Bose polaron by the repulsion between bosons, in the framework of the Gross-Pitaevskii theory. It constitutes an important contribution since this stabilization was absent in most theories. The paper is very clearly written and the results are physically sound. All of the journal's acceptance criteria are met, and I think it duly deserves publication once the authors have clarified a few points and considered the following suggestions.

Reply: We appreciate the referee’s positive evaluation of our manuscript and recommendation for publication. We also thank the referee for their effort put into this detailed and thorough report which helped us to further improve our manuscript and clarify some important points according to the referee’s suggestions.

Referee: Here are some points I would like to clarify: 1) The authors should specify more clearly the range of validity of their theory. In Section III.5, they state "our local Gross-Pitaevskii theory is reliable as long as the potential range R is not so short that the assumption of a slowly varying potential is violated". With respect to which length scale should the range R not be too short? Later they hint that the theory could be extended to reff << aBB, so I suppose the relevant length is aBB. If so, why is it the relevant length scale? This should be discussed more explicitly.

Reply: We thank the referee for this comment that is also in line with the other referee reports highlighting that our discussion of the range of validity of our approach was not sufficiently clear. In our work we follow the work by Chen, Prokofiev, and Svistunov (PRA 98, 041602 (2018)) where they show that indeed aBB is the relevant length scale and the GPE approach remains applicable if aBB<<R, where R is the range of the potential determining the impurity-boson interaction. In our work we ensure that even at the smallest ranges shown in Fig. 5, the range is at least three times larger than aBB, hence the classical field approximation is valid. We have modified the associated discussion in our work.

Referee: 2) The authors investigate the universality of their results by considering two different potentials: a Gaussian and an exponential potential. These two potentials are relatively similar. To get a more convincing evidence of universality, it would be interesting to consider a markedly different potential. For instance, if it is not too much work for the authors, it would be interesting (and more physically motivated) to use a van der Waals potential, such as a Lennard-Jones potential. This is just a suggestion.

Reply: We appreciate the referee’s comment and suggestion. While it is in principle possible to perform the calculation also for van der Waals potentials, these typically support a large number of bound states which in our approach (that yields a bound on the ground state energy of the system) will lead to the occupation of the lowest state of this potential. On the other hand, the scattering length determined by the scattering threshold physics. In this situation, the polaron physics would be determined by the lowest bounds states in the interaction potential which in turn would be independent of the scattering length. Indeed, one would need to adapt the GPE approach to essentially project out all but the weakest bound state to obtain results relevant for the typical adiabatic preparation of polaron states close to a Feshbach resonance. This is however, while highly interesting, beyond the scope of our work.

Referee: 3) In Fig. 5, it would be instructive to plot the impurity-boson dimer energy for comparison. If possible, it would also be interesting to plot results for vanishing n0. Intuitively, that limit should bridge the mean-field regime and the dimer energy.

Reply: We chose to plot energies in units of the finite density n0 in the system. Hence, to make such a plot as suggested by the referee one should use a different set of absolute units which are density independent. In our work we have chosen density units in order to highlight the universality of the predictions for experimental realizations at different densities.

Referee: 4) In Fig. 5, it is stated that the polaron energy at negative scattering length is universal for reff>= xi. Is it a numerical observation for that particular calculation or a more general statement? The authors should clarify the generality of this statement.

Reply: This statement was made with regard to the specific calculation shown in Fig. 5c. We have now modified the manuscript accordingly.

Referee: 5) It is pointed out that a deep Gaussian potential can have several bound states, which makes the polaron energy non-unique. At first, it is not clear why an emphasis is put on the Gaussian potential. It would seem that the exponential potential would equally have several bound states if sufficiently deep. I presume that in the range of effective ranges considered, the required depth of the potentials is such that a second bound state appears for the Gaussian potential, but not for the exponential potential. Is that right? Somehow this discussion could be clarified. It might help to specify either in the text or on the curves of Fig 5(d) the number of bound states supported by each potential.

Reply: The referee is of course correct that also an exponential potential could support an arbitrary number of bound states if sufficiently deep. Their intuition is also right that for the effective ranges considered for the Gaussian potential there are indeed two bound states while for the exponential one finds only one. We now clarify this in the manuscript.

Referee: I have a few additional comments: 6) The authors explain that the stabilization of the Bose polaron comes from the repulsion between the bosons. However, it has been shown in Ref. [55] that even for non-interacting bosons, the interaction between the impurity and bosons can have a stabilizing effect. The authors may want to comment on this additional effect and whether it is present in their theory (presumably not) and what it would take to include this effect.

Reply: In Ref. [55] it was nicely shown how the exchange of a closed-channel dimer can yield effectively a constraint of the occupation of bosons close to the impurity, thus having a similar effect as a boson repulsion. The inclusion of this mechanism is not straightforward to include in the approach which assumes density-density interactions between impurity and bosons.

Referee: 7) In section II.A, the authors state that their ansatz "is based on a coherent ansatz". I think the product form they consider is actually a Fock state, and not a coherent state.

Reply: We thank the referee for pointing this out. We now corrected that statement and made it consistent. We now clarify that for large particle number both in the Fock and coherent state ansatz all relevant observables become indistinguishable.

Referee: 8) In Section VI, the authors state "However, when the coherent state ansatz or higher-order excitation extensions of the Chevy ansatz [41, 55] are applied to the truncated Hamiltonian (4) they lead to the aforementioned, spurious divergence of the ground-state energy". I do not think that there is any spurious divergence of the ground-state energy in Refs. [41][55]. Perhaps the authors meant to refer to Ref. [30]?

Reply: In this case we aimed to refer to a divergence due to the so-called Thomas collapse where the energy of the lowest Efimov state goes to infinity in the zero-range limit at fixed scattering length. We agree that our discussion was not sufficiently transparent in this regard and we reformulated the corresponding text in the manuscript.

Reply to Report 2

Referee: Strengths: A relatively simple solution to the Bose polaron problem that removes the divergence in the energy.

Weaknesses: The GP theory is unlikely to be valid for short-range impurity-boson potentials. This should be more clearly discussed.

This paper certainly meets the requirements for publication. It investigates the behavior of an impurity in a BEC, a topic of intense interest in quantum gases, and it provides a way to remove the divergence in the ground-state energy that has plagued many previous approaches, such as Ref [30]. It also presents a simple classical picture of how bosons effectively screen the impurity and thus reduce the strength of the attractive impurity potential. The main points that require more discussion are the validity of the classical-field approach and how the current work fits with other theoretical approaches.

Reply: We thank the referee for the positive evaluation of our manuscript and the helpful comments in particular with regard to the validity of the approach and comparison to others works.

Referee: Specifically, I would like the authors to address the following questions/comments: 1. What is the precise condition for the classical field (GP) approach to be valid? I would expect that the range R has to be sufficiently large but does it need to be larger that the healing length or just aBB ?

Reply: We thank the referee for this question which in fact is similar to the one raised by Referee 1 (Point 1). As we outline there, we now explain in the text that R needs to be larger than aBB as shown in previous work by Chen et al. [PRA 98, 041602 (2018)].

Referee: 2. How does that the current approach differ from that in Ref. [38]? The authors of Ref [38] appear to use the same GP equation, yet claim a larger regime of validity, i.e., R~aBB .

Reply: Reference [38] is based on the classical field equations (GPE). However, we apply this approach under more conservative assumptions (R>>aBB). Moreover, we also discuss the case of aIB>0 and present concrete numerical results for this case where the repulsive interactions between bosons competes with bound state formation.

Referee: 3. Does the approach in the paper capture the perturbative results in the limit of weak boson-impurity coupling? For instance, it is known that the GP approach in Ref [43] does not recover it properly, while the coherent-state approach in Ref. [30] does.

Reply: Our approach corresponds to the one in [43], but extended to aIB>0 where the two-body bound state appears. The perturbative limit R simeq aIB << aBB is an interesting one which, however, goes beyond our assumption of R>>aBB discussed above.

Referee: 4. I was confused about why it was claimed that higher-order extensions of the Chevy ansatz lead to a spurious divergence. In Ref. [41] the dependence on a high-momentum cutoff is due to Efimov physics, which is a physical effect that is also present in QMC calculations.

Reply: In this case we aimed to refer to a divergence due to the so-called Thomas collapse where the energy of the lowest Efimov state goes to infinitely in the zero-range limit at fixed scattering length. The inclusion of three-body correlations in higher-order Chevy ansätze naturally capture the Efimov effect and with it the Thomas collapse that implies the divergence of the Efimov trimer energy in the strict limit of contact interactions. We agree that our discussion was not sufficiently transparent in this regard and we reformulated the corresponding text.

Referee: 5. Can the approach be extended to non-zero temperature?

Reply: Yes, this should be possible in principle and one might consider following the ideas outlined in the work by Dzsotjan, Schmidt & Fleischhauer, Dynamical variational approach to Bose polarons at finite temperatures, Physical Review Letters, 124, 223401.

Referee: Requested changes: 1. A clear discussion of the validity of the GP theory in the paper.

Reply: See answer to point 1. We modified the text accordingly.

Referee: 2. More comparison with previous work. In particular, how does the theory compare with the QMC calculations for an ionic impurity in Ref. [44]?

Reply: We modified the text to include a new reference to Chen et al. PRA 98, 041602(R) (2018) and to clarify the relation to works employing expansions in the number of medium excitations. The question about the comparison to the case of an ionic impurity is certainly an interesting one. There, however, a longer range potential ~1/r^4 is of relevance that is beyond the focus of this work.

Referee: 3. Can the authors comment on the limit of a zero-range potential? Since this has been used in other works including QMC calculations.

Reply: This is indeed an intriguing question. This limit is outside of the validity of our approach. There are however several models that allow to treat this limit, such as the use of the Chevy approach and its extensions or e.g. the non-local GPE equation [Drescher et al., PRR 2, 032011(R) (2020)].

Referee: 4. The authors should also comment more on Ref [55], which shows that an infinitely heavy impurity can trap an infinite number of bosons but still have a lower bound on the ground-state energy. An infinite number of bosons trapped by a short-range potential was also shown in Chen et al., PRA 98, 041602(R) (2018).

Reply: Indeed Chen et al. show that a deep potential well with a bound state can lead to a collapse of the BEC with an infinite number of bosons attracted to the well while the ground state energy remains still finite. However, this particular ground state is extremely dilute and the particle number of the Bose polaron cloud grows sublinear with the volume as V^1/3. In contrast, we consider high density systems with a much larger particle density than accommodated by this collapse solution. Hence the solution shown in Fig. 2 approaches the background density at positive chemical potential and contains a finite number of bosons attracted to the impurity. While the effect discussed by Chen et al. is similar to Ref. 55 in terms of a diverging number of particles at a finite energy, they originate from distinct microscopic models (Ref. 55 considers a two-channel model) so that the comparison is not straightforward.

Reply to Report 3

Referee: In the present work the authors elaborate on the stabilization mechanism of Bose polarons occurring for strong local attractions. Indeed, it is known that using conventional approaches, e.g. the Fröhlich model, an instability emerges in this strongly attractive interaction regime and the bosons of the medium accumulate towards the impurity. To tackle this problem a variational type approach is constructed that operates beyond the standard Bogoliubov approximation including the local boson repulsion which acts against the impurity’s attractive potential excluding binding. As argued this additive term is able to stabilize the Bose polaron in the vicinity of the respective scattering resonance. For instance, it is demonstrated that the polaron energy is bounded from below across the resonance while the polaron dressing cloud remains sizable. Moreover, at resonance, the polaron energy shows a universal behavior on the effective range and the impurity contact is finite exhibiting a peaked structure at positive scattering lengths.

I find the results and ideas presented intriguing with direct experimental relevance and most likely will inspire similar theoretical investigations in the future. The manuscript is also well written. However, I have some questions regarding the findings and also the applicability of the used method. Thus, if the authors provide convincing answers to my comments, summarized below, and perform the respective minor revisions then I would certainly recommend this work for publication in Sci post.

Reply: We appreciate the referee’s thorough review of our work, recommendation for publication after minor revision and the many useful comments and questions.

Referee: 1) I have some conceptual questions regarding the applicability of the used model, e.g. described by Eqs. (1) or (2). a) As far as I understand it holds for both mobile and immobile impurities (Figure 3(b)). However, I am not able to clearly judge its applicability for finite impurity masses, meaning that in the latter case impurity-medium correlations is possible to become important. Can the authors briefly comment on this issue?

Reply: The approach is indeed applicable to both mobile and immobile impurities. With the reduced mass appearing in the units used in Fig. 3b the effect of the finite impurity mass is included. Certainly impurity-bath correlations are important and these are included in the wave function phi that (in the comoving frame) relates to the impurity-boson pair correlation function.

Referee: b) Is it possible the present formalism to be extended for larger impurity concentrations, e.g. more than a single one? Would then its reliability be restricted?

Reply: The mechanism of stabilization by boson repulsion certainly applies also to the case of a dilute impurity gas. More subtle is the question about the direct extension of the computational approach itself to the multi-impurity case. The Lee-Low-Pines (LLP) transformation allows for the elimination of the impurity operator. In the case of multiple impurities one would, however, only be able to eliminate a single impurity coordinate with the others remaining, so that it becomes an interesting question how the problem can still be mapped onto a purely bosonic one.

Referee: c) Can it be extended to account also for long-range interactions e.g. dipoles? I imagine that the emergent physics especially in c) will be fundamentally different than the one described in the manuscript but I wonder what is the authors opinion here.

Reply: For long-range impurity-bath interactions one of the challenges will be that the scattering physics cannot easily be described by a scattering length. It is indeed a very interesting question for future studies how long-range interactions compete with the short-range repulsion between bosons.

Referee: 2) On page 3, first paragraph of the left column, it is stated that “For bosons in the vicinity ... leads to induced interactions between bosonic particles ...”. What type of induced interactions are meant here? Also, are they always attractive or is it possible to engineer also repulsive ones?

Reply: The induced interactions are due to the impurity kinetic energy term after the LLP transformation. It has the form of a current-current interaction between bosons. This kinetic energy term competes with the attractive potential V_IB. In order to optimize the energy of the bosons in this attractive potential, the kinetic energy term will have to be minimized. This is achieved by the formation of complex boson correlations.

Referee: 3) Is it possible to compare or translate the modified quasiparticle dispersion to the one obtained for particle-balanced mixtures e.g. introduced in Phys. Rev. Lett. 117, 100401 (2016) by Petrov? I can understand that this might be well beyond the scope of the present work but if the authors are able to provide any relevant hint would be extremely useful not only for advancing the impact of their work but also for the community.

Reply: We did not attempt to calculate the polaron quasiparticle dispersion in terms, e.g., of an effective mass correction. This can be achieved in our approach by investigating the ground state energy as function of p_0. In our work we solely focused on the case of p_0=0.

Referee: 4) On page 3, first paragraph of the right column, I would suggest besides Ref. [40] to include also Refs. New J. Phys. 21, 103026 (2019), New J. Phys. 22, 043007 (2020), that also refer to the one-dimensional case and elaborate on the effect of attractive impurity-boson interactions while relying on a variational treatment.

Reply: We thank the referee for their suggestion and have included the references in the revised manuscript.

Referee: 5) It is also not clear to me how the present model accounts for impurity-medium correlations. Please comment. I would expect that such effects would become particularly important in the nonequilibrium polaron dynamics which the authors claim to be a next step to apply their formalism.

Reply: These correlations are certainly the most important aspect of our work. They are shown explicitly in Fig. 2, which displays the impurity-boson density-density correlation determined by the wave function phi: because of the LLP transformation this is directly related to the impurity-medium correlation. Certainly these correlations are important also in the non-equilibrium dynamics as discussed e.g. in Drescher et al. PRR 2020.

Referee: 6) Regarding the dynamical instability appearing in the GPE. Is it related to the presence of Bogoliubov modes with complex eigenfrequencies leading to the instability of the corresponding stationary solution of the GPE solution? Or is it one of the so-called thermodynamic ones as introduced e.g. by Castin in arXiv:cond-mat/0105058?

Reply: The dynamical instability is discussed in the literature typically in the context of the time evolution starting from a homogeneous BEC. In our work we focus on the ground state properties originating from the deformation of the BEC so that the question of a dynamical instability does not directly arise in our discussion.

Referee: 7) I wonder what is the behavior of the polaron residue in the interaction region of the bound state? Is it finite? If no, how then the polaron picture should be interpreted?

Reply: This is an interesting point. Indeed, we did not calculate the quasiparticle weight in our work but we expect it (for finite boson repulsion) to remain finite, despite being small.

Referee: 8) I am bit perplexed regarding the results shown in figure 3 (b). On the one hand in the caption it is argued that the polaron binding energy is presented when equal mass of boson and impurity is considered and for two different gas parameters. However, in the text it is written that “...we present the energy for a mobile impurity of arbitrary mass...”. Please comment. Also, why the authors do not show the polaron binding energy for a heavier impurity and one of the scattering lengths used for the equal mass case? This will enable to judge also the impact of the mass on the binding energy.

Reply: The GPE solutions are applicable for arbitrary impurity-boson mass ratios. With the proper choice of units (that include masses) a universal scaling curve can be derived (Fig. 3a) that applies to any mass ratio. In Fig. 3b we then choose the units typically used in the literature (“density units”). For these units the mass ratio has to be explicitly specified and in our work we focus on the mass-balanced case in order to complement previous literature that focused on the infinite impurity mass limit.

---

## Round 2 · List of Changes

discuss validity of GPE approach
expanded discussion of multiple bound states in deep potentials
added missing linear impurity coupling in Eq. (4)
added remark on coherent and product states
added Refs. [32,33,35,37,44-46,48]

---

## Round 3 · Referee Report · Anonymous (Referee 1) · 2021-12-1

Report

  1. I thank the authors for removing the confusion about "coherent states" in their manuscript.

  2. Concerning my suggestion to use a third model potential to confirm the claimed universality, I am surprised that the authors repeatedly declined to do so. It is very easy to do once a numerical code for minimizing Eq. (8) has been written, which the authors have done. I did the numerical calculation myself, and found that model potentials with a van der Waals tail reproduce the results of the manuscript (see attached figure). I am now convinced that the universality claimed by the authors on the basis of only two examples is likely to be correct. Again, I invite the authors to do this calculation themselves and include it in their article to strengthen their claim.

  3. There seems to be a typo in Eq. (8). The $\xi^2$ factor inside the integral should be removed, and a $\xi^3$ factor should be added in front of the integral. Could the authors confirm this point? Also, the notation implies that E in Eq. (7) and (8) are the same, but as far as I understand, the authors have (duly) subtracted the condensate energy -$\mu n_0 /2$, which is divergent for increasing volume. This should be made clear in the notations as well as in the text.

  4. I am confused about the units of Eq. (17). According to Eq. (8) (once the typo is corrected), the energy is proportional to $\mu n_0 \xi^3$ and to a dimensionless function of the dimensionless potential parameters $a/\xi$ and $r_{\rm eff}/\xi$. At unitarity, it reduces to a function of only $r_{\rm eff}/\xi$, which for $r_{\rm eff}/\xi \gtrsim 0.2$ has the linear form $A + B\times (r_{\rm eff}/\xi)$, where A and B are found numerically to be about -5 and -9. It follows that Eq. (17) should read: $E/E_n = A n_0^{1/3}\xi + B n_0^{1/3}r_{\rm eff}$ In other words, there should be a factor $n_0^{1/3}\xi = (8\pi n_0^{1/3} a_{BB} m_{\rm red}/m_B)^{-1/2}$ in the first term of the right-hand side. Can the authors confirm whether this is the case?

  5. The comparisons with the works of Refs. [42] and [59] are very confusing.

5.1 About the comparison with Ref. [42] (Massignan et al): Eq. (11) of Ref. [42] states that at unitarity, the polaron energy for $R << \xi$ is: $E = - (3\pi n_0 \xi/m_B) (R/\xi)^{1/3}$ where $R = 0.557 r_{\rm eff}$ for a square well potential. Expressing it in units of $E_n = n_0^{2/3}/(2m_{\rm red}) \approx n_0^{2/3}/(2m_B)$, one finds: $E = - E_n 6 \pi (\frac{R}{8\pi a_{BB}})^{1/3}$ The authors state that this formula is consistent with their results for the particular value $r_{\rm eff} = 2 a_{BB}$. Where does this value come from? I do not understand why the authors expect an agreement for this particular value. It is also unclear how the value -5 is obtained, as well as how the horizontal coordinate of the green star in Fig. 5b is determined. My understanding is that Ref. [42] and the present manuscript essentially use the same GP theory. However, Ref. [42] applies this theory in the limit $r_{\rm eff}/\xi \ll 1$, whereas the present manuscript considers the theory for $r_{\rm eff}/\xi \gtrsim 0.2$. At unitarity, the two papers obtain results in separate regimes, namely: $E/(\mu n_0 \xi^3) = -3\pi (0.557 y)^{1/3}$ for Ref. [42] ($y\ll1$) $E/(\mu n_0 \xi^3) = -5 -9 y$ for the present manuscript ($y\gtrsim0.2$) where $y = r_{\rm eff}/\xi$. Plotting these two functions of y (see dotted and dashed curves in the attached figure) shows that they do not cross. One can imagine that they connect to each other in the intermediate regime, but it should be observed from Fig. 1 or Ref. [42] that numerical results match the first function only for $y\lesssim0.002$. Therefore, the mentioned value -5 and star in Fig. 5 look very misleading.

5.2 About the comparison with Ref. [59] (Levinsen et al). Eq. (6) of Ref. [59] states: $E = -E_n 2 f( n_0^{1/3} a_{BB})$ where the function $f$ is found by QMC calculations to follow the law: $f(x) = -0.36 ln( 0.019 x)$ This looks inconsistent with the results of the present manuscript. However, the authors state that this formula is consistent with theirs for the particular value $x = n_0^{1/3} a_{BB} = 0.04$. Again, where does this value come from? If one considers the extrapolation to zero range of the linear law found in this manuscript, one finds (with the missing factor I mentioned in point 4.): $f(x) = -2.6 /\sqrt{8\pi x}$ which does coincide with the previous formula at $x=0.04$, but this is simply a coincidental point for two completely different laws.

As an aside, Ref. [42] also mentions that $f(x) = \sqrt{\pi/4x}$ for the coherent ansatz of Ref. [30]. That formula is consistent with the result of the present manuscript, if multiplied factor of about 1.77. Perhaps this is worth mentioning.

I would be grateful to authors for clarifying these points.

Requested changes

  1. Confirm/Clarify the points mentioned in the report
  2. Add the van der Waals potential data to Fig. 5.
  3. Remove the star in Fig. 5, unless the authors can give a justification. To make a comparison with Ref. [42], I suggest the authors to display the curve corresponding to the formula $E/(\mu n_0 \xi^3) = -3\pi (0.557 y)^{1/3}$ obtained from Ref. [42].

Attachment

---

## Round 3 · Author Response

Reply to Report 2

Referee:
 The authors have gone some way towards addressing the comments of the referees, particularly regarding the regime of validity of their theory. However, there are still some remaining points that I would like to see addressed before publication: - If the regime of validity corresponds to R≫aBB, then presumably this is why the results of the GP theory do not agree with the QMC results in [Phys. Rev. Lett. 127, 033401 (2021)] ? This should be mentioned in the manuscript.

Reply: The comparison with the Monte Carlo results is an interesting point. Contrary to the expectation of the referee, the results show very good agreement for the parameter set for which both works make their predictions, specifically aBB=0.04 n^(-1/3) and r_eff->0. In this caseQMC predicts E=-2.6 n^(2/3)/mB in perfect agreement with the short-range extrapolation of our results in Eq. (17) for the infinite-mass case. We now mention this in the manuscript, along with the reference to the QMC work.

Referee:
- I do not think that the perturbative limit of weak boson-impurity coupling violates the assumption of R≫aBB, as claimed by the authors. Rather, one requires aIB to be small compared to the BEC healing length, and this can be satisfied by taking the boson density to be sufficiently low, without making any assumptions about the range of the potential. Therefore, if the GP theory does not match the perturbative weak-coupling results, this suggests that the theory cannot accurately capture the low-density limit. The authors should at least comment on this in the manuscript.

Reply: In the low-density limit mentioned by the referee, the impurity potential of range R with xi >> aIB ~ R >> aBB becomes effectively a contact potential on the scale of the healing length. In our manuscript we already discuss below Fig. 5(b) and in Section VI that this limit is not well described by the standard GPE: while for an attractive impurity potential with aIB < 0 a Chevy ansatz might be sufficient, for repulsive aIB > 0 the nonlocal GPE [40] is capable of describing the large polaron cloud around a short-range impurity.

Reply to Report 1

Referee:
 1. In their response, the authors argue that atomic interactions typically support a large number of bound states, which makes it too difficult to use a van der Waals potential in their work. The authors could have nonetheless considered a shallow van der Waals potential with only one bound state (such as that of helium) to compare with the Gaussian/Exponential potential and check their claimed universality. This would not be particularly more difficult than a Gaussian or exponential potential with one bound state, and would significantly strengthen the manuscript. If the authors do not do this simple check in the present work, it will need to be be done in another publication (if it is ever done). I think this is a good opportunity for the authors, but again I leave it to their discretion.

Reply: We thank the referee for this suggestion; the systematic analysis of the range dependence of the ground-state energy for different model potentials is a worthwhile topic for further research.

Referee:
 2. I find it regrettable that the authors insist on referring to their ansatz as a "coherent state". Technically, as I pointed out in my first report, they introduce it as a Fock state, not as a coherent state, and it is equivalent with a coherent state only in the limit of large numbers of particles. Moreover, the coherent nature (superposition of states with different numbers of particles) does not play any role in this work. I think terms like "Fock state" or "mean-field" or "Gross-Pitaevskii" would be much more easily understood. Moreover, they oppose their work to the coherent ansatz of Ref. [30,36], which truly is a coherent ansatz where the superposition of number states plays a crucial role (here, the number of excitations). Naming both approaches as coherent ansatz may cause confusion to the readers.

Reply: Our work is concerned with the thermodynamic limit of large particle number where the product and coherent states have indistinguishable density expectation values, and the fluctuations of the particle number become small compared to the total particle number. In this limit, the states used in our work and in Refs. [30,36] become equivalent, with one written in real space and the other in momentum space. Instead, the main difference is that the Hamiltonian is bounded from below in our work. To avoid confusion, we removed references to coherent states.

---

## Round 3 · List of Changes

List of changes: - compared our prediction for the ground-state energy with the recent QMC result in J. Levinsen et al., Phys. Rev. Lett. 127, 033401 (2021), which now appears as reference [59]. - refer to Eq. (6) as product state.

---

## Round 4 · Referee Report · Anonymous (Referee 1) · 2022-7-6

Report

I thank the authors for their efforts to improve their manuscript, in particular the calculation with the Lennard-Jones potential which further demonstrates the universality of their results. I am satisfied with all the replies and changes. The manuscript constitutes a nice piece of work and I believe it is now fit for publication in SciPost.

---

## Round 4 · Author Response

The referee writes:

  1. I thank the authors for removing the confusion about "coherent states" in their manuscript.

Our response: We thank the referee again for having pointed this out.

The referee writes:

  1. Concerning my suggestion to use a third model potential to confirm the claimed universality, I am surprised that the authors repeatedly declined to do so. It is very easy to do once a numerical code for minimizing Eq. (8) has been written, which the authors have done. I did the numerical calculation myself, and found that model potentials with a van der Waals tail reproduce the results of the manuscript (see attached figure). I am now convinced that the universality claimed by the authors on the basis of only two examples is likely to be correct. Again, I invite the authors to do this calculation themselves and include it in their article to strengthen their claim.

Our response: We recognise the referee’s effort in providing an excellent report on our manuscript and appreciate their concern with regards to testing the claim of universality further. In addition to the data shown by the referee we have thus now also evaluated the case of a physical Lennard-Jones potential given by:

$$ V_{\rm LJ}(r)=\frac{\lambda R^{10}}{r^{12}} - \frac{R^4}{r^6} . $$
Here, $\lambda$ and $R$ are chosen to produce the desired scattering length $a$ and effective range $r_{\rm eff}$, respectively. In contrast to the cases considered previously, this potential features also a strong repulsive contribution, thus allowing to test the degree of universality of our results further. Again, we find good agreement with our previous results. We added this data now also to Fig. 5 as suggested by the referee. We expect that for very large ranges the repulsive core limits the applicability of universality.

The referee writes:

  1. There seems to be a typo in Eq. (8). The $\xi^2$ factor inside the integral should be removed, and a $\xi^3$ factor should be added in front of the integral. Could the authors confirm this point? Also, the notation implies that $E$ in Eq. (7) and (8) are the same, but as far as I understand, the authors have (duly) subtracted the condensate energy $-\mu n_0 /2$, which is divergent for increasing volume. This should be made clear in the notations as well as in the text.

Our response: There was a typo in Eq. (8), and we thank the referee for spotting this. Indeed a bracket was set incorrectly which we have now corrected (in our convention $u$ has units of length). Moreover, we have subtracted the energy $-\mu n_0 /2$ of the unperturbed BEC and we now mention this explicitly in the text above Eq. (8).

The referee writes:

  1. I am confused about the units of Eq. (17). According to Eq. (8) (once the typo is corrected), the energy is proportional to $\mu n_0 \xi^3$ and to a dimensionless function of the dimensionless potential parameters $a/\xi$ and $r_{\rm eff}/\xi$. At unitarity, it reduces to a function of only $r_{\rm eff}/\xi$, which for $r_{\rm eff}/\xi \gtrsim 0.2$ has the linear form $A + B\times (r_{\rm eff}/\xi)$, where $A$ and $B$ are found numerically to be about -5 and -9. It follows that Eq. (17) should read:
    $$ E/E_n = A n_0^{1/3}\xi + B n_0^{1/3}r_{\rm eff} $$
    In other words, there should be a factor $n_0^{1/3}\xi = (8\pi n_0^{1/3} a_{BB} m_{\rm red}/m_B)^{-1/2}$ in the first term of the right-hand side. Can the authors confirm whether this is the case?

Our response: The referee is correct about this missing factor which we have now included in the new version of the manuscript. We thank the referee for spotting this!

The referee writes: 5. The comparisons with the works of Refs. [42] and [59] are very confusing.\ 5.1 About the comparison with Ref. [42] (Massignan et al):\ Eq. (11) of Ref. [42] states that at unitarity, the polaron energy for $R \ll \xi$ is:

$$ E = - (3\pi n_0 \xi/m_B) (R/\xi)^{1/3} $$
where $R = 0.557 r_{\rm eff}$ for a square well potential. Expressing it in units of $E_n = n_0^{2/3}/(2m_{\rm red}) \approx n_0^{2/3}/(2m_B)$, one finds:
$$ E = - E_n 6 \pi (\frac{R}{8\pi a_{BB}})^{1/3} $$
The authors state that this formula is consistent with their results for the particular value $r_{\rm eff} = 2 a_{BB}$. Where does this value come from? I do not understand why the authors expect an agreement for this particular value. It is also unclear how the value -5 is obtained, as well as how the horizontal coordinate of the green star in Fig. 5b is determined.\ My understanding is that Ref. [42] and the present manuscript essentially use the same GP theory. However, Ref. [42] applies this theory in the limit $r_{\rm eff}/\xi \ll 1$, whereas the present manuscript considers the theory for $r_{\rm eff}/\xi \gtrsim 0.2$. At unitarity, the two papers obtain results in separate regimes, namely:
$$ E/(\mu n_0 \xi^3) = -3\pi (0.557 y)^{1/3} \,\,\textrm{for Ref. [42]}\,\, (y\ll1)\ E/(\mu n_0 \xi^3) = -5 -9 y \,\,\text{for the present manuscript}\,\, (y\gtrsim0.2) $$
where $y = r_{\rm eff}/\xi$. Plotting these two functions of y (see dotted and dashed curves in the attached figure) shows that they do not cross. One can imagine that they connect to each other in the intermediate regime, but it should be observed from Fig. 1 or Ref. [42] that numerical results match the first function only for $y\lesssim0.002$. Therefore, the mentioned value -5 and star in Fig. 5 look very misleading.

Our response: Ref. [42] includes a Fig. 1 that shows the polaron energy as function of the effective range. Also in this Figure a nearly linear regime is visible at large range. With the comparison shown in Fig. 5 we wanted to convey that our predictions for the energy agree in this regime (no surprise since the referee states correctly that both we and Ref. [42] apply GPE theory). We did not make or wanted to imply any statements about the scaling predictions in [42], and indeed our comparison was made for a value of the effective range well outside the regime where Ref. 42 predicts their 1/3 scaling. We acknowledge that a comparison might have been more confusing than helpful and thus we now opted to take up the referee’s comment and we removed this explicit comparison altogether from the manuscript (in particular in terms of the green star depicted in Fig. 5).

As a side remark ---also coming back to the referee’s second comment--- one might actually regard the agreement in energy with Ref. [42] as further supporting our claim of universality since Ref. [42] employs yet a different microscopic potential compared to our work.

The referee writes: 5.2 About the comparison with Ref. [59] (Levinsen et al).\ Eq. (6) of Ref. [59] states:

$$ E = -E_n 2 f( n_0^{1/3} a_{BB}) $$
where the function $f$ is found by QMC calculations to follow the law: $f(x) = -0.36 ln( 0.019 x)$. This looks inconsistent with the results of the present manuscript.\ However, the authors state that this formula is consistent with theirs for the particular value $x = n_0^{1/3} a_{BB} = 0.04$. Again, where does this value come from?\ If one considers the extrapolation to zero range of the linear law found in this manuscript, one finds (with the missing factor I mentioned in point 4.): $f(x) = -2.6 /\sqrt{8\pi x}$ which does coincide with the previous formula at $x=0.04$, but this is simply a coincidental point for two completely different laws.\ As an aside, Ref. [42] also mentions that $f(x) = \sqrt{\pi/4x}$ for the coherent ansatz of Ref. [30]. That formula is consistent with the result of the present manuscript, if multiplied factor of about 1.77. Perhaps this is worth mentioning.\ I would be grateful to authors for clarifying these points.

Our response: Here again we only aimed at comparing our energy values to other literature to check for overall consistency. Our comparison to Ref. [59] highlighted that although predicted scalings are very different, actual energies might still be very much comparable for typical experimental parameters. However, as the referee’s comment makes clear a comparison for just a single point (in order to test the energy) causes more confusion than it is helpful and thus we again opted to remove this discussion from the manuscript.\ Concerning the referee’s comments on the $\sqrt{1/x}$ scaling found in Ref. [30]: In this case we refrain from making explicit comments or comparisons since Ref. [30], by applying the Bogoliubov approximation, really represents a result for a fundamentally different model than the one considered in the present work.

Requested changes by the referee: 1. Confirm/Clarify the points mentioned in the report.\ 2. Add the van der Waals potential data to Fig. 5.\ 3. Remove the star in Fig. 5, unless the authors can give a justification. To make a comparison with Ref. [42], I suggest the authors to display the curve corresponding to the formula $E/(\mu n_0 \xi^3) = -3\pi (0.557 y)^{1/3}$ obtained from Ref. [42].

Our response: 1. We thank the referee for bringing up these points. We have addressed these in the report above. \ 2. We have added additional data to Fig. 5 obtained for a Lennard-Jones potential including the van der Waals tail, further supporting the claim of universality.\ 3. Taking into account the referee’s valuable input, in order to unambiguously avoid any confusion or misunderstanding, we have opted to remove the comparison with Ref. [42] and the star from the figure as discussed in our reply above.

---

## Round 4 · List of Changes

1. Included new data for a Lennard-Jones potential in Fig. 5b, further supporting universality. Removed the green star from the figure which lead to confusion.
  2. Fixed typos in Eqs. (8) and (17).
  3. Stated explicitly that the polaron energy is measured relative to the energy of the unperturbed BEC.

---

## Editorial Decision

published